# Microbiota-Dependent and -Independent Production of L-Dopa in the Gut of *Daphnia magna*

Rehab El-Shehawy,[a] Sandra Luecke-Johansson,[a] Anton Ribbenstedt,[a] Elena Gorokhova[a]

[a]Department of Environmental Science, Stockholm University, Stockholm, Sweden

Sandra Luecke-Johansson and Anton Ribbenstedt contributed equally to this work.

**ABSTRACT** Host-microbiome interactions are essential for the physiological and ecological performance of the host, yet these interactions are challenging to identify. Neurotransmitters are commonly implicated in these interactions, but we know very little about the mechanisms of their involvement, especially in invertebrates. Here, we report a peripheral catecholamine (CA) pathway involving the gut microbiome of the model species *Daphnia magna*. We demonstrate the following: (i) tyrosine hydroxylase and Dopa (3,4-dihydroxyphenylalanine) decarboxylase enzymes are present in the gut wall; (ii) Dopa decarboxylase gene is expressed in the gut by the host, and its expression follows the molt cycle peaking after ecdysis; (iii) biologically active L-Dopa, but not dopamine, is present in the gut lumen; (iv) gut bacteria produce L-Dopa in a concentration-dependent manner when provided L-tyrosine as a substrate. Impinging on gut bacteria involvement in host physiology and ecologically relevant traits, we suggest L-Dopa as a communication agent in the host-microbiome interactions in daphnids and, possibly, other crustaceans.

**IMPORTANCE** Neurotransmitters are commonly implicated in host-microbiome communication, yet the molecular mechanisms of this communication remain largely elusive. We present novel evidence linking the gut microbiome to host development and growth via neurotransmitter L-Dopa in *Daphnia,* the established model species in ecology and evolution. We found that both *Daphnia* and its gut microbiome contribute to the synthesis of the L-Dopa in the gut. We also identified a peripheral pathway in the gut wall, with a molt stage-dependent dopamine synthesis, linking the gut microbiome to the daphnid development and growth. These findings suggest a central role of L-Dopa in the bidirectional communication between the animal host and its gut bacteria and translating into the ecologically important host traits suitable for subsequent testing of causality by experimental studies.

**KEYWORDS** *Daphnia magna*, L-Dopa, interkingdom communication, host-microbiome interactions, *Daphnia*, dopamine synthesis, gut microbiome, molt cycle and development, peripheral pathways for neurotransmitters

The role of gut bacteria in regulating host homeostasis and feedback between the microbiome and its host is a hot topic in current evolutionary, ecological, and biomedical studies. Interkingdom communication is the bidirectional flow of signals between the host neurophysiological system and its microbiome, with neurotransmitters acting as signaling molecules in the host-microbiome metabolic axis. Bacteria recognize and produce common vertebrate neurotransmitters (1–3), such as catecholamines (CA)—L-Dopa (L-3,4-dihydroxyphenylalanine), dopamine, epinephrine, and norepinephrine (4). L-Dopa is the precursor for dopamine. Although L-Dopa itself is a neurotransmitter/modulator with receptors in the central and peripheral nervous system (5), its role as a signaling molecule *per se* in interkingdom communication is unclear. Mutualistic bacteria have been found

Address correspondence to Elena Gorokhova, elena.gorokhova@aces.su.se.

responding to L-Dopa variation, e.g., *Enterobacteriaceae* and *Pseudomonadaceae* increase their growth *in vitro* when supplemented with L-Dopa (6). Moreover, several bacterial taxa produce L-Dopa *in vivo*, and the microbial enzymes tyrosinase, tyrosine phenyl lyase, and *p*-hydroxyphenylacetate 3-hydroxylase are exploited in the biotechnological L-Dopa production (7). Thus, bacteria recognize, respond, and produce L-Dopa.

Animal guts are rich in CA and harbor commensal bacteria, especially in the epithelium-associated biofilms. Therefore, active CA-mediated communication between the gut microbiota and the animal host is likely to occur (4). Of particular challenge is understanding molecular mechanisms and pathways of the host-microbiome interactions in organisms of differing complexity, moving away from mammalian studies to other relevant model species in ecology and evolution research. Using invertebrate models to understand the precise mechanisms of the host-microbiome metabolic axis has been advocated given the complexity of host and microbial metabolism and the diversity of the mammalian microbiome (8). However, our knowledge of microbiome-mediated regulation of host development, immunity, homeostasis, and behavior progresses beyond the established model animals is limited, and most of what we know about host-microbiome interactions is based on vertebrate models. In evolutionary ancient neuronal circuits of the invertebrates, sensory cells have receptors for neurotransmitters, neuropeptides, and other signaling molecules in the neural membrane, which may still have been incompletely genetically individualized compared to more complex vertebrates (9). Dopamine, octopamine, and acetylcholine are examples of neurotransmitters present in both vertebrate and invertebrate nervous systems, and symbiotic bacteria have been shown to affect some neurotransmitters. The octopamine signaling pathway, for example, is used by commensal bacteria to manipulate the host sensory decision in *Caenorhabditis elegans* (10) and male aggression in *Drosophila melanogaster* (8).

The branchiopod crustaceans in the genus *Daphnia* are invertebrate models broadly used in ecology and evolution studies, including host-microbiome interactions (11–16). The central nervous system in the water fleas and other lower crustaceans consists of the brain, compound eye, optic ganglia, and thoracic nervous system (17–19). The neuropils, neurosecretory somata, and neurotransmitter-producing neurons visualized with immunostaining included histaminergic (17), peptidergic (20), and dopaminergic neurons (21). However, we know much less about the peripheral nervous system in these models, especially the gut innervation, and no sensory neurons in the gut have been visualized in *Daphnia* (18). We also know that *Daphnia* microbiome modulates the life cycle, including growth, reproduction, and tolerance to environmental stressors (11, 13–15), yet the molecular basis of the interactions affecting these ecologically and evolutionary relevant traits is poorly understood.

Here, we hypothesized that CA are involved in host-microbiome interactions in *Daphnia magna*, and these interactions occur at the gut-lumen interface (Fig. 1). The hypothesis was tested using a series of experiments with *D. magna* to detect L-Dopa and dopamine in the gut lumen, localize the enzymes catalyzing the first steps of the CA pathway, follow the decarboxylation of L-Dopa to dopamine by the host during the molt cycle, and evaluate the gut microbiota ability to produce L-Dopa. Collectively, these data were used to understand L-Dopa production in the gut by the host and its microbiome.

## RESULTS

**Localization of peripheral CA pathway in the gut.** Tyrosine hydroxylase (TH) and Dopa decarboxylase (DDC) enzymes were localized in the daphnid midgut whole mount (Fig. 2a) using immunohistochemistry (experiment 1 [Exp I]), suggesting a peripheral CA pathway in the gut wall. The TH presence (Fig. 2b and c) indicates hydroxylation of L-tyrosine to L-Dopa, which is the first and critical limiting step of the CA pathway, whereas the DDC presence (Fig. 2d and e) indicates decarboxylation of L-Dopa to dopamine, the second step (Fig. 1). Thus, the occurrence of TH and DDC in the gut

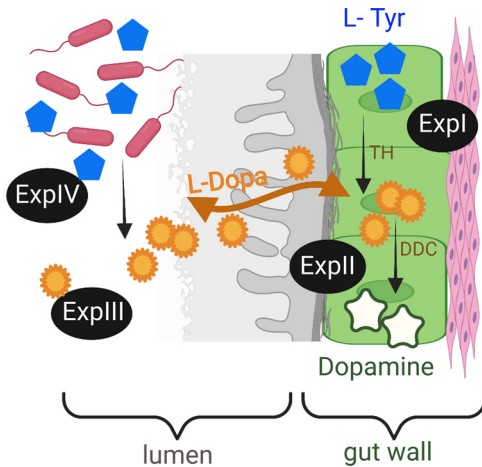

**FIG 1** Conceptual diagram presenting the hypothesis and the associated experiments (Exp I to Exp IV). The first step in the eukaryotic CA pathway is the hydroxylation of ʟ-tyrosine (ʟ-Tyr) to ʟ-Dopa via tyrosine hydroxylase (TH). It is the rate-limiting step followed by the decarboxylation of ʟ-Dopa to dopamine by Dopa decarboxylase (DDC). ʟ-Dopa is of dual origin and is an information signal in a putative bidirectional communication between *Daphnia* and its gut bacteria in the lumen.

wall, including the microvillous layer, suggests a local dopamine synthesis associated with the gut wall, including epithelium.

**Dynamics of *Ddc* expression during the molt cycle.** In Exp II, we used a quantitative PCR (qPCR) assay to measure the *Ddc* gene expression in the daphnid gut during the molt cycle with ecdysis as a reference point (i.e., postmolt, intermolt, and premolt). The *Ddc* gene was expressed in all samples tested (Fig. 2f), indicating that the DDC synthesis is continuous. Moreover, there was a significant association between the *Ddc* expression and the daphnid molt stage, with the highest values observed in the postmolt animals (pairwise comparison with the Holm-Šídák test; $F_{4,5} = 10.34$, $P < 0.02$; Fig. 2f).

**ʟ-Dopa in the lumen.** In Exp III, we used liquid chromatography−high-resolution mass spectrometry (LC-HRMS) to identify CA in *Daphnia* lumen samples prepared from the dissected guts. Peaks of the exact mass corresponding to ʟ-Dopa (<3 dPPM) were detected in all samples, and spectrum fragmentation revealed a similarity of 0.905 with

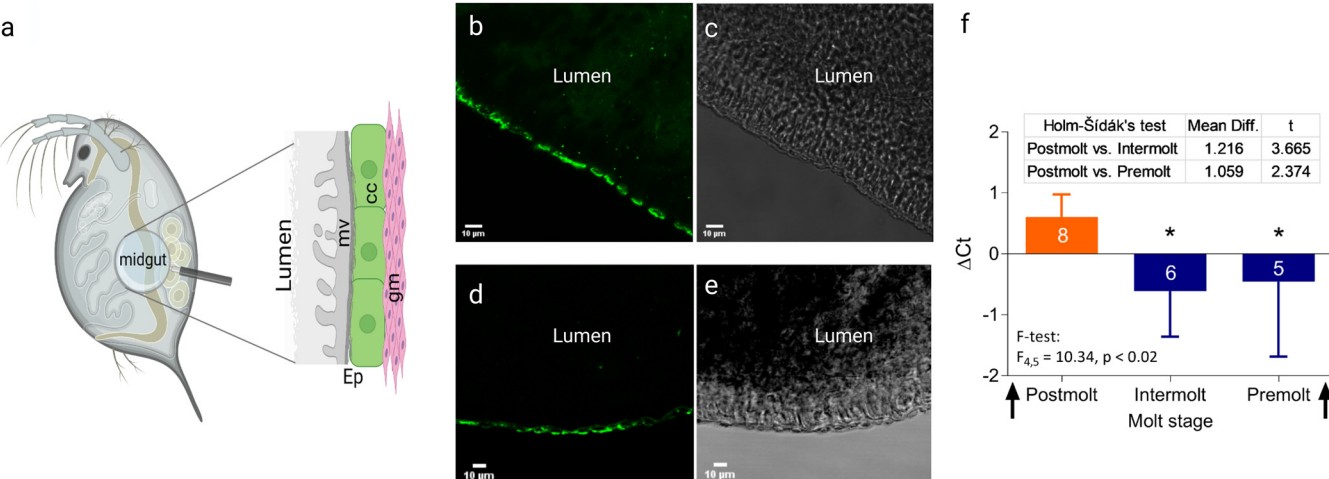

**FIG 2** ʟ-Dopa production in *Daphnia magna* gut. (a) Schematic structure of the midgut (37) analyzed with immunostaining (mv, microvillous layer; cc, columnar cells; gm, gut musculature; Ep, gut epithelium comprised of mv and cc). (b and d) Immunofluorescence localization of TH in the midgut epithelial cells; (c and e) Immunofluorescence localization of DDC in the midgut epithelial cells; the same region as for TH but in a different individual. Labeling with Alexa Fluorophore 488 is shown on the left, and the light microscopy of the same specimen is shown on the right; no labeling appeared in the unlabeled control (not shown). (f) The significant upregulation of *Ddc* gene expression in *Daphnia* gut observed during the postmolt decreasing during the intermolt and premolt; arrows indicate ecdysis. The data are shown as z scores (mean ± standard deviation [SD]; number of observations are shown on the bars for each stage) normalized to the cycle-specific average values for each of the replicate experimental runs.

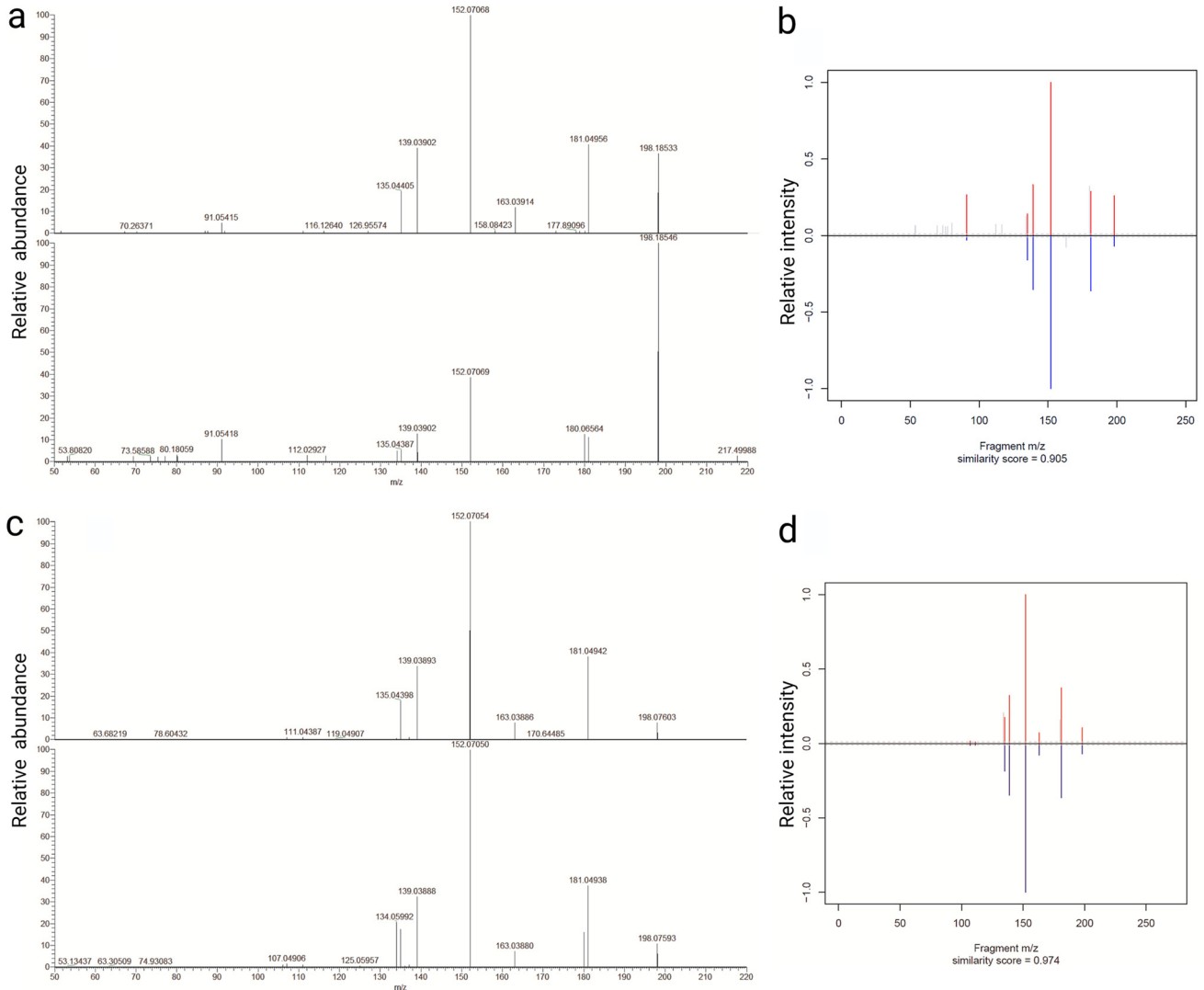

**FIG 3** Presence of L-Dopa in lumen and its production by the enriched gut microbiome of *D. magna*. (a) Identification of L-Dopa in *Daphnia* lumen by comparing fragmentation pattern of the native standard (top spectrum) to the endogenous L-Dopa in the lumen (bottom spectrum). (b) Similarity score from MSMSsim analysis for peaks with exact mass of L-Dopa from the *Daphnia* lumen versus the standard. (c) Identification of L-Dopa produced by the microbiome *in vitro* (2 mM L-tyrosine treatment) by comparing fragmentation pattern of the native L-Dopa standard (top graph) and bacteria-produced L-Dopa (bottom graph). (d) MSMSsim similarity score for peaks with the exact mass of the bacterium-produced L-Dopa and the standard.

a native L-Dopa standard (Fig. 3a and b), allowing positive L-Dopa identification (22). Dopamine was not detected in these samples at a limit of detection (LOD) of 0.47 pg/µl; therefore, it was either below the detection limit or conjugated and biologically inactive (23).

**Production of L-Dopa by gut bacteria.** The L-Dopa detected in the lumen could be produced by the host (as shown in Exp I) and/or the microbiota that utilizes foodborne L-tyrosine for L-Dopa synthesis (7). To evaluate whether the daphnid microbiota is capable of L-Dopa production, we conducted an enrichment experiment *in vitro* by amending gut bacteria *in vitro* with L-tyrosine at 0 to 3 mM concentration range (Exp IV). Bacterial growth responded to the enrichment in a dose-dependent manner (based on optical density at 600 nm [$OD_{600}$] dynamics; see Fig. S1 in the supplemental material). Spectral fragmentation of the L-Dopa produced by the bacteria revealed spectral similarities above 0.949 with the native L-Dopa standard at ≥2 mM L-tyrosine (Fig. 3c and d). Therefore, L-Dopa detected in the *Daphnia* gut lumen can originate from both the host and its microbiota.

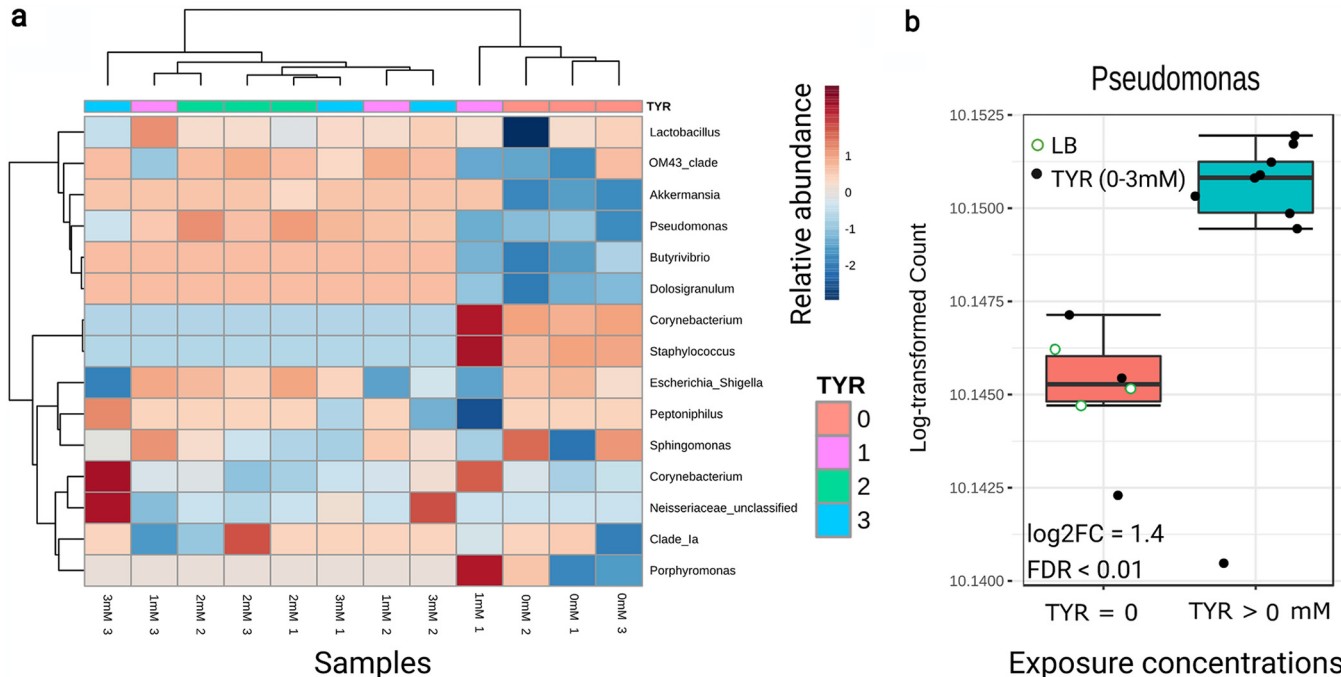

**FIG 4** Identification of taxa producing L-Dopa. (a) Heat-map with cluster analysis at the genus level (>0.2%) for relative abundance. (b) Results of the differential abundance analysis for significantly upregulated taxa in the L-tyrosine treatments showing that *Pseudomonas* was significantly associated with the L-tyrosine exposure. The samples were grouped to show abundances from 0 and LB (TYR = 0) and 1 to 3 mM (TYR > 0 mM) treatments. See Fig. S3 in the supplemental material for the composition of the bacterial community in the experiment.

**Bacterial taxa associated with L-Dopa production.** The 16S rRNA gene sequencing of the bacterial cultures inoculated using daphnid gut microbiota and grown under different levels of L-tyrosine enrichment (Exp IV) resulted in a total of 647,891 high-quality filtered reads, with a mean read depth per sample of 43,192 sequences and a total amplicon sequence variant (ASV) number of 120 (49 with ≥2 counts). The bacterial taxa detected in the L-tyrosine incubations (Fig. S3) overlapped with the *D. magna* gut microbiota reported earlier for this clone (16, 24). The differential abundance analysis showed that *Pseudomonas*, *Akkermansia*, and *Butyrivibrio* were upregulated in the L-tyrosine exposure, with significant upregulation in *Pseudomonas* (Fig. 4 and Fig. S4). According to the principal coordinate analysis (PCoA), the communities that were developing without L-tyrosine clustered closely together, which separated them from those exposed to any L-tyrosine (1, 2, and 3 mM) along the first principal coordinate (PC) axis (Fig. 5). Once the multivariate homogeneity was confirmed (homogeneity of multivariate dispersion [PERMDISP]; $F$ value, 0.05841; $P$ value, 0.8128), a permutation test was performed which detected significant differences between the groups driven by L-tyrosine exposure (permutational multivariate analysis of variance [PERMANOVA]; $F$ value, 8.497; $R^2$, 0.3959; $P$ value < 0.007).

## DISCUSSION

We demonstrated for the first time that the neurotransmitter L-Dopa is produced in the gut jointly by *Daphnia* and its microbiome and used for dopamine synthesis in concert with the molt cycle progression of the host. Moreover, immunostaining of TH and DDC in the gut wall suggests a peripheral CA pathway in *D. magna*. To the best of our knowledge, no cells in the gastrointestinal tract that contain dopamine and express components of dopamine signaling pathways, including enzymes and specific dopamine receptors and transporters, have been reported in *Daphnia*. In decapod crustaceans and insects (25–28), dopamine and other neurotransmitters are present in the gut innervation, contributing to gut motility regulation, but there are only a few studies on the neuroendocrine cells associated with gut lining and their functioning. Also,

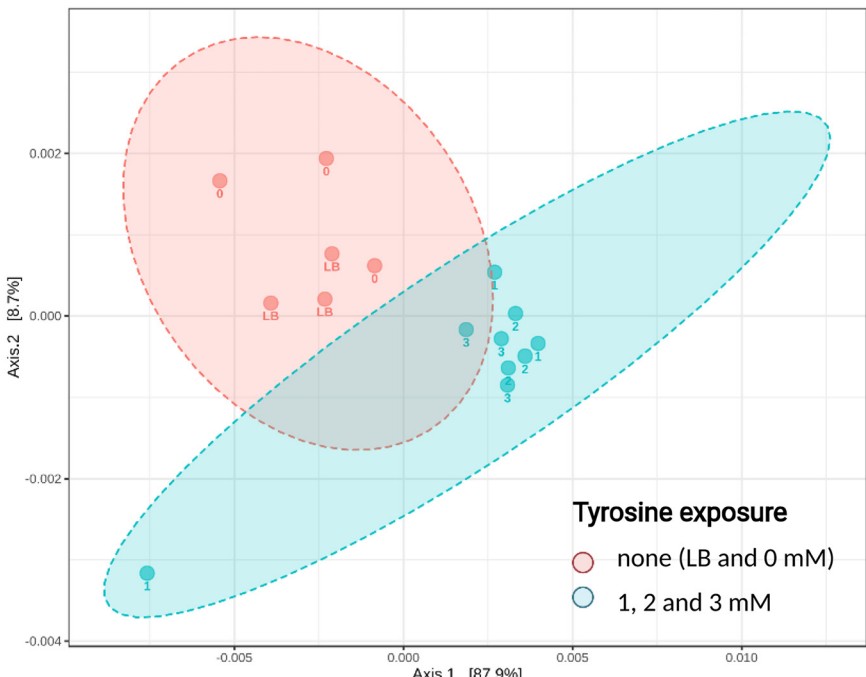

**FIG 5** Principal coordinate analysis (PCoA) based on Bray-Curtis dissimilarity metrics, showing the distance in the bacterial communities between the treatments. The samples were grouped to show abundances from 0 and LB (TYR = 0) and 1 to 3 mM (TYR > 0 mM) treatments. See Fig. S4 for the relative abundance of *Pseudomonas* in these samples. log2FC, $\log_2$ fold change.

whether these neurotransmitters are uniquely active in the enteric nervous system or whether they also act in the central nervous system is not yet fully understood. Notably, in *Drosophila* larvae, only 6% of DDC activity is associated with the brain, the rest occurs in the epidermis (29). Moreover, enteroendocrine cells account for 5 to 10% of the midgut epithelial cells in flies (30). In the gastrointestinal tract and other mesenteric organs of vertebrates, dopamine production is substantial (31, 32), and gut epithelial cells contain DDC and receptors for L-Dopa with both endogenous L-Dopa and lumen L-Dopa involved in dopamine synthesis (33).

The central nervous system in the water fleas has been mapped using immunostaining. However, we know much less about the peripheral nervous system in these model species, especially the gut innervation, and no sensory neurons in the gut have been visualized in *Daphnia* (17, 18). Histaminergic somata have been identified along the neuropils extended from the dorsal and ventral cords surrounding the gut, but none were seen in the periphery reaching the gut wall (17) as shown for *Artemia* shrimp (34). Our immunohistochemical analysis (Fig. 2a to e) has revealed relatively homogenous labeling pattern along the gut wall. Given the observed staining of the microvillous layer, we suggest that TH and DDC detected in our study are produced by the gut lining. However, this does not exclude the possibility that both neurotransmitter-producing somata and dopaminergic fibers similar to the serotoninergic fibers associated with the gut musculature in insects (35) are involved.

The release mechanisms for L-Dopa and dopamine in the cuticle of *Daphnia* or other crustaceans and their physiological triggers are poorly understood. We found free unconjugated and thus biologically active L-Dopa in the *Daphnia* lumen, but no dopamine, which supports the suggested pathway of the dual origin of lumen L-Dopa functioning as an information signal in a bidirectional communication between the animal and its gut bacteria (Fig. 1). Biosynthesis of amino acid transporters by *Daphnia* microbiota was found in metagenome-assembled genomes, supporting this pathway

(36). Moreover, the differences between the L-Dopa and dopamine transport systems may also contribute. Both molecules are transported using systems adapted to their chemical structure. In neurons, dopamine is transported via the specialized vesicular monoamine transporter (VMAT) or dopamine transporter (DAT) (37, 38), while L-Dopa is transported via the large neutral amino acid transporter (LNAA) system (39), a heterodimeric membrane transport protein that preferentially transports branched-chain and aromatic amino acids. Thus, a lack of a specialized transport system may explain why we detected L-Dopa but not dopamine in the lumen.

The *Ddc* gene is evolutionary highly conserved across taxa (40, 41). We followed the expression of *Ddc* in the gut and found that it follows the molt cycle, peaking after ecdysis. As DDC is an enzyme found in metazoans (40), whereas bacteria use other pathways to transform L-Dopa to dopamine (42), the observed expression pattern reflects daphnid *Ddc* gene activity. According to the qPCR results and confocal microscopy, transcription and biosynthesis of DDC occur in the gut wall, which is in line with a correlation between epidermal *Ddc* expression and the protein activity reported in insects, with no translational modification (40). Dopamine and its derivatives are essential for arthropod physiology, including gut motility, cuticle formation, and sclerotization of the integument during the postmolt (35, 41), and thus involved in the basic functions, i.e., feeding, ontogenetic development, behavior, wound healing, and protection against pathogens (40). As feeding and cuticle formation are crucial for growth and development and tightly regulated by the molting hormone ecdysone, *Ddc* is also under tissue-specific and hormonal control (43). In line with our *Ddc* expression results, epidermal *Ddc* activity in *Drosophila* was reported to peak before embryo hatching, during larval-larval molting, pupariation, and adult eclosion (40, 43). DDC is a rate-limiting enzyme for cuticle sclerotization, and mutation loss of *Ddc* in *Drosophila* is lethal for embryos (38). Therefore, the cuticle formation in *Daphnia* foregut and hindgut (44) is a possible target for the *Ddc* gene expression and the associated DDC production peaking shortly after ecdysis. However, other targets and processes following the molt cycle can be involved.

Besides facilitating digestion, gut microflora participates in host metabolism and behavior through their ability to produce, recognize, and modulate eukaryotic neurotransmitters and other information signals (2). We found that the gut microbiota of *Daphnia* is capable of L-Dopa production *in vitro* following enrichment with L-tyrosine. Moreover, exposure to L-tyrosine caused a significant shift in bacterial community structure (Fig. 5), with some taxa, such as the endogenous *Pseudomonas* spp. being significantly upregulated in a concentration-dependent manner (Fig. 4 and Fig. S4). In line with this, several bacterial taxa have been reported to produce L-Dopa, including *Pseudomonas* spp. (45), and several bacterial enzymes for L-Dopa production have been identified (7). Thus, we suggest that *Daphnia* and its gut bacteria are engaged in host-microbiome communication using L-Dopa as a signaling molecule. Our findings that L-Dopa is produced by both the gut epithelium and the microbiota and reports showing that bacteria recognize and respond to L-Dopa (6) support this hypothesis. In turn, as a grazer, *Daphnia* can use chemosensation to modulate feeding behavior and food uptake in response to amino acids present in the feeding environment (46). Tyrosine and its precursor phenylalanine were found to stimulate *D. magna* mandible movements by 43% and 34%, respectively (46), which would provide an increased supply of this amino acid to the microbiota and the substrate for L-Dopa synthesis. One can speculate that the host genotype might affect the bidirectional communication via, for instance, responding to different CA molecules or hosting specific CA-producing taxa.

In summary, TH and DDC are present in the *Daphnia* gut wall, and both the host and its microbiota contribute to the free unconjugated L-Dopa in the gut (Fig. 1). Therefore, L-Dopa is a putative agent for host-microbiome communication in daphnids and, perhaps, other invertebrates translating into the ecologically important host traits. In invertebrates, gut bacteria affect epithelium development (37, 38), modulate growth

factor signaling, and gut stem cell activity (47). The fact that *Ddc* is always expressed in *Daphnia*, peaking after ecdysis, suggests continuous dopamine synthesis, its possible involvement in active feeding during the postmolt, cuticle formation, and ontogenetic development as well as behavioral traits and phenotypic plasticity at large (48). Therefore, by contributing to ʟ-Dopa production, the gut bacteria may affect dopamine synthesis, host performance, behavior, and adaptation. In turn, by modulating the amino acids and, particularly, tyrosine intake, the host might benefit by regulating microbial ʟ-Dopa synthesis, and this regulatory capacity may vary both inter- and intraspecifically.

Our work provides new insights into the molecular mechanism(s) by which lower crustaceans communicate and interact with their microbiome and increase our fundamental knowledge about the role of ʟ-Dopa, not as a dopamine precursor as conventionally assumed, but as an agent of interkingdom communication in crustacean ecophysiology and regulation of ecological traits. Further work will identify the biochemical, physiological, and ecological context for the host-microbiome interactions conveyed by lumen ʟ-Dopa and explore its commonality across phylogenies.

## MATERIALS AND METHODS

**Experimental design.** The experimental design focused on identifying the key reactions in the CA pathway, i.e., the hydroxylation of ʟ-tyrosine to ʟ-Dopa and the decarboxylation of ʟ-Dopa to dopamine, and the evaluation of the possible contribution of ʟ-Dopa of bacterial origin to the total pool of ʟ-Dopa in the *Daphnia* gut. Altogether, four experiments (Exp I to IV; Fig. 1) were conducted. Exp I was conducted to localize the CA pathway in the gut wall using immunohistochemistry to confirm hydroxylation of ʟ-tyrosine, which is responsible for hydroxylation of ʟ-tyrosine to ʟ-Dopa, and Dopa decarboxylase (DDC) carrying out the decarboxylation of ʟ-Dopa to dopamine. Exp II was conducted to obtain time series of the expression of *Ddc* gene in the gut over the molt cycle using qPCR assay. Exp II was conducted to detect L-Dopa and dopamine in *Daphnia* lumen using LC-HRMS. Exp IV was conducted to evaluate ʟ-Dopa production by the gut microbiota using enriched cultures of gut bacteria supplemented with ʟ-tyrosine to stimulate the growth of taxa capable of utilizing this substrate and measuring ʟ-Dopa with LC-HRMS.

***Daphnia magna* culture and standardization of test animals.** All experiments were conducted with a single clone of *Daphnia magna* Straus (Cladocera, Branchiopoda; clone V, obtained from the Federal Environment Agency, Berlin, Germany). The animals are cultured under standard conditions (49) in M7 medium in groups of 12 individuals liter$^{-1}$ at 20 ± 2°C and 16-h/8-h light/dark cycle. The food, a mixture of green algae, *Pseudokirchneriella subcapitata* and *Scenedesmus spicatus*, was provided three times a week, and the medium was changed once a week.

Under these conditions, the molt cycle duration in instars III to VI (the developmental stages used in this study) was 1.8 to 3.2 days. The animals used for immunostaining in Exp I were standardized with regard to their molt and embryo development stages: only Instars V-VI in the premolt stage carrying black-eyed embryos were used. The molt stage chronology was established in pilot experiments and found to follow the published schedules well (50, 51). In Exp II, this chronology was used to assign the test animals to three main stages—postmolt, intermolt, and premolt with midpoints 28%, 60%, and 85% of the molt cycle duration, respectively.

In daphnids, the digestive tract consists of an ectodermal foregut, an endodermal midgut, and an ectodermal hindgut. Before the gut sample collections for all four experiments, the daphnids were allowed to swim in sterile M7 medium for 10 min as a washing step and transferred to a sterile microscopy slide. Next, the gut tube without the ceca (hereafter referred to as a gut sample) was dissected in each individual using sterile forceps and dissection needles.

**Localization of TH and DDC in the gut wall.** In Exp I, immunohistochemistry was used to localize tyrosine hydroxylase (TH) in the daphnid gut whole mount; only premolt animals were used for this analysis (*n* = 7). For primary antibodies, we used rabbit polyclonal antibodies against TH (50997, Nordic BioSite; previously used to identify dopaminergic neurons in *D. magna* [21]) and goat polyclonal antibodies against Dopa decarboxylase (DDC) (catalog no. AF3564; Novus Biologicals). Cross-absorbed Alexa Fluor 488-labeled goat antirabbit IG (H+L) (catalog no. A32731; ThermoFisher Scientific) and cross-absorbed Alexa Fluor 488-labeled donkey anti-goat IgG (H+L) (catalog no. A11055; ThermoFisher Scientific) were used as secondary antibodies. Negative controls without the primary antibodies or using IgG isotypes (goat IgG [catalog no. NB410-28088; Novus Biologicals] and rabbit IgG [catalog no. NBP2-2489; Novus Biologicals]) at the same dilution as the primary antibodies were used.

Immunolabeling was carried out following the established methods (17, 18) with some modifications. In brief, the fixation of the dissected guts was carried out overnight in 4% paraformaldehyde. Samples were incubated with 5% bovine serum albumin (BSA) overnight (Super blocker; ThermoFisher Scientific) before applying primary antibodies. Incubation with primary antibodies was conducted for 48 h in the presence of 1% BSA (Super blocker; ThermoFisher Scientific). Incubation with secondary antibodies was conducted for 24 h in the presence of 1% BSA. The concentrations of antibodies were 1:2,000 for both primary and secondary antibodies. The samples were rinsed three times for 10 min each

time, followed by three times for 1 h each time, and then once overnight using phosphate-buffered saline with 0.5% Triton X-100 (Sigma-Aldrich) (PBS-TX). All incubation and washing steps were done at 4°C on an orbital shaker in the dark. Single images and Z-scans were taken using Zeiss confocal microscope 710 equipped with Aragon laser at 20× and 40× magnification and the manufacturer software. Images were processed using ImageJ2 (52); no image manipulation was applied.

**RNA extraction, reverse transcription-quantitative PCR (RT-qPCR), and gene expression analysis.** In Exp II, synchronized cohorts were used to generate samples for premolt, intermolt, and postmolt stages. Total RNA was extracted from the gut samples (7 to 10 guts/sample) using the RNeasy minikit (Qiagen) and the on-column DNase I treatment (catalog no. 79254; Qiagen) according to the manufacturer's instructions with additional in-tube DNase I treatment (AMPD1; Sigma-Aldrich). We used G3PDH (glyceraldehyde 3-phosphate dehydrogenase) as a housekeeping gene, which has a stable expression in *Daphnia* (53, 54). The primers for *Ddc* and *G3PDH* assays were adopted from Campos et al. (54) (see Table S1 in the supplemental material); both primers were used to control for any residual DNA contamination. The RNA was reverse transcribed using the High-Capacity RNA-to-cDNA kit (ThermoFisher Scientific) following the manufacturer's instructions. The qPCR assays were conducted with Applied Biosystems StepOne real-time PCR system using QuantiNova SYBR green (Qiagen) as follows: cDNA (1 $\mu$l, equivalent to 5 ng), forward and reverse primers (1 $\mu$M), 2× QuantiNova SYBR green PCR Master Mix (5 $\mu$l), QuantiNova ROX reference dye (1 $\mu$l), and DNA/RNA-free water (2 $\mu$l; Sigma-Aldrich) at 95°C for 2 min, 35 cycles, with 1 cycle consisting of 20 s at 95°C and 20 s at 60°C for annealing and data acquisition. A melt curve was generated after each run to ensure the reaction specificity. The $\Delta C_T$ values were calculated (55) to estimate the relative *Ddc* gene expression.

***In vitro* L-Dopa synthesis by *Daphnia* gut microbiome.** In Exp IV, preenrichment was carried out by incubating whole guts in LB for 48 h at 24°C with shaking. The overnight LB cultures were used to inoculate (10% inoculum [vol/vol]) test mixtures containing L-tyrosine at concentrations of 0, 1, 2 and 3 mM in M7 medium; each concentration treatment was in triplicate. The inoculated cultures were then incubated for 24 h at 24°C with shaking. At the end of the incubation time, the culture density was measured at $OD_{600}$. Sample preparation for LC-HRMS was conducted as follows: 100 $\mu$l of culture was transferred into a tube containing 300 $\mu$l of an ice-cold solvent mix of acetonitrile (ACN) and MilliQ mixtures, 85:15, with the addition of 0.1% (vol/vol) formic acid (FA), followed by quick vortexing and centrifugation at 4°C for 10 min. After centrifugation, the supernatant was transferred to LC-grade vials (ThermoFisher Scientific) and stored at −80°C until analysis.

**LC-HRMS.** In Exp III, to obtain lumen samples for LC-HRMS, the dissected guts (20 guts/sample, four samples) were transferred to microplate wells containing 50 $\mu$l of ice-cold M7 medium/well, and the plate was kept on ice thereafter. The guts were allowed to leak out their contents into the ice-cold M7 by applying gentle stirring. After 1 to 2 min, 50 $\mu$l of each lumen content suspended in M7 was transferred to an Eppendorf tube containing 150 $\mu$l of ice-cold solvent mixture. The lumen samples were then sonicated for 15 min and centrifuged at 14,000 rpm for 10 min at 4°C. The supernatant was then transferred into 320-$\mu$l insert LC-grade glass vials and stored at −80°C until analysis.

The LC-HRMS analysis was conducted to identify the total (Exp III) and bacterium-produced (Exp IV) L-Dopa. All analytical standards were purchased from Sigma-Aldrich and were of analytical grades (98 to 99% purity). Methanol (MeOH), acetonitrile (ACN), and FA were also purchased from Sigma-Aldrich and were of the highest purity (98 to 99%). MilliQ water was produced in-house using a Milli-Q Integral 3 (LC-Pak polisher, Millipak express 40 filters, 0.22 $\mu$m, Merck) for a final organic content of <3 ppb. Native standards of L-Dopa and dopamine were prepared by dissolving the respective powders in MeOH and MilliQ mixtures (80:20), with the addition of 1% (vol/vol) FA, which was then diluted with pure methanol to obtain a 0.1% FA in >98% MeOH solution.

Subsequently, these solutions were diluted with a 0.1% FA ACN solution to reach the final concentrations of 1.29 and 0.26 $\mu$M for the two native standards used for spiking and identification. The standard was prepared in a 1.5-ml glass vial by mixing 800 $\mu$l of the standard solution with 200 $\mu$l MilliQ and 200 $\mu$l ACN.

Instrumental analysis was carried out through injection on the high-performance liquid chromatography (HPLC)-HRMS system (Ultimate 3000 and Q Exactive HF Orbitrap; ThermoFisher Scientific) with an adapted version of the electrospray ionization settings as in Ribbenstedt et al. (56) (positive ionization, 3,700 V; sheath gas, 30; auxiliary gas, 10; sweep gas, 0; S-lens RF 50; capillary and auxiliary gas heater temperature, 350°C) with data-independent MS2 acquisition [full scan: 120,000 (120k) resolution (res); max IT 100 ms; AGC target 3e6; scan range, 70 to 1,000 *m/z*; ddMS2: 30k res; max IT 100 ms; AGC target 1e5; loop count 3; TopN 3; isolation window 0.4 *m/z*; (N)CE 30; ddSettings: Min AGC target 1.00e1; Apex trigger 1 to 5 s; charge exclusion, 5 to 8 and >8; Excl.isot. "On"; Dyn.excl. 2.0 s; If idle "Do not pick others"]. The system was equipped with an in-line filter (0.5 $\mu$m) before the precolumn-fitted hydrophilic interaction liquid chromatography (HILIC) column (Both BEH Amide, 1.7 $\mu$m, 2.1 × 5 mm and 2.1 × 150 mm, Waters, USA). The exact mass of L-Dopa was also added to the inclusion list to guarantee MS2 acquisition. To rule out matrix effects on the retention time (RT) of L-Dopa in the sample matrices and to evaluate intensity drift over the injection sequence, quality control (QC) samples were prepared by fortifying several already injected replicates of each sample matrix (i.e., lumen and gut microbiome grown on L-tyrosine) through the addition of 10 $\mu$l of three of the calibration mixtures into one replicate each. This way all RT deviations were accounted for and sequence intensity drift was shown to be 8% over the injections.

All chromatograms were integrated and quantified in XCalibur 3.063 (ThermoFisher Scientific). MS2 spectra were extracted to mzML format using MSConverter (51) (Peak pick settings: Prefer vendor "check," MS lvls "1-"; Subset settings: Scan number "", Scan time "", Mz win. 0.0-198.10) and compared

with the R-script MSMSsim (57) with identities being considered confirmed at a confidence level 1, according to the Schymanski scale (22), when similarity scores > 0.9 and when retention times (RT) in spiked samples matched the samples.

**16S rRNA gene sequencing.** In Exp IV, the bacterial communities grown at different L-tyrosine concentrations, including controls, were used for next generation sequencing (NGS) analysis by sequencing the 16S rRNA gene using the V3-V4 hypervariable region and primers 341F (5′-CCTACGGGNGGCWGCAG-3′) and 805R (5′-GACTACHVGGGTATCTAATCC-3′) (58). Genomic DNA was extracted from all samples using DNeasy PowerBiofilm extraction kit (Qiagen) and then purified with AMPure XP beads (Beckman Coulter) following the manufacturer's instructions. After the purification, the DNA concentrations were quantified using Quant-iT PicoGreen double-stranded DNA (dsDNA) assay kit (ThermoFisher Scientific) and Tecan Ultra 384 SpectroFluorometer (PerkinElmer). Quality control was performed on an Agilent 2100 BioAnalyzer using a high-sensitivity DNA chip.

The library preparation and sequencing were conducted at LC Sciences/LC Bio (Houston, TX, USA) using the Illumina NovaSeq 6000 platform (2 × 250 bp paired-end) following the manufacturer's instructions (Illumina, San Diego, CA, USA). Blanks with nondetectable DNA levels and no PCR product were not used for sequencing. Paired-end reads were assigned to samples based on their unique barcode, truncated by removing the barcode and primer sequences, and merged using FLASH to 400 bp. *Fqtrim* (v0.94; https://ccb.jhu.edu/software/fqtrim/index.shtml) was used to filter raw tags for harvesting high-quality clean tags, and the *Vsearch* software (v2.3.4; VSEARCH, GitHub, https://github.com/torognes/vsearch) was applied to filter the chimeric sequences. After dereplication with DADA2, taxonomy was assigned using SILVA classifier (release v. 132, confidence level > 0.7) with q2-feature-classifier (v2019.7; https://github.com/QIIME2/q2-feature-classifier), a QIIME 2 (https://qiime2.org) plug-in for taxonomy classification of marker-gene sequences (59). Sequences have been deposited with links to BioProject accession number PRJNA694094.

**Data analysis. (i) Evaluation of the differences in the *Ddc* gene expression among the molt stages.** As the gene expression data were collected using different cohorts on three occasions, we normalized data by z-score transformation, using mean value and standard deviation for each cohort, to standardize data across the experimental runs. The normalized data were independent of the absolute variation between the individuals from different cohorts and used to compare groups (postmolt, intermolt, and premolt) by the pairwise multiple-comparison Holm-Šídák test with multiplicity-adjusted *P* values (60). The null hypothesis was rejected with a probability of error $\alpha < 0.05$.

**(ii) Microbial community structure and identification of taxa responding to L-tyrosine.** In the sequence analysis, data filtering was applied using a minimum count of 4 and 20% prevalence to remove low-quality or uninformative features. Due to the moderate variability in the sequence libraries, the data were not rarefied for diversity analysis. Rarefaction curves and Zhang-Huang's coverage estimator (see Fig. S2 in the supplemental material) were calculated from ASV abundances using functions supplied by the *vegan* and *entropart* R packages.

To visualize the differences in the bacterial community structure, log-transformed counts were used for a heat-map with cluster analysis at the genus level (>0.2%). The R-package *edgeR* was used to identify differentially abundant bacterial taxa based on the false discovery rate (FDR)-corrected *P* values ($\alpha = 0.05$, FDR = 1%) that were associated with controls and L-tyrosine incubations; for normalization, a trimmed mean of M-values (TMM) between each sample pair was used. The principal coordinate analysis (PCoA) with Bray-Curtis dissimilarity index was used to visualize differences in bacterial community composition between the treatments with L-tyrosine concentration of ≥1 mM and those not exposed to L-tyrosine (LB-grown bacteria and controls). Differences in the community structure at the genus level were tested by permutational multivariate analysis of variance (PERMANOVA); Bray-Curtis dissimilarity was used as variance-stabilizing transformation. Multivariate homogeneity of treatment dispersion was assessed using the *betadisper* function in the *vegan* R package. PHYML in Geneious Prime 2019.2.3 was used to reconstruct the phylogenetic tree using 16S rRNA gene sequences of all treatments. All figures were prepared using BioRender.

**Data availability.** Sequences have been deposited with links to BioProject accession number PRJNA694094.

## SUPPLEMENTAL MATERIAL

Supplemental material is available online only.
**FIG S1**, TIF file, 0.02 MB.
**FIG S2**, TIF file, 0.2 MB.
**FIG S3**, TIF file, 0.2 MB.
**FIG S4**, TIF file, 0.5 MB.
**TABLE S1**, DOCX file, 0.01 MB.

## ACKNOWLEDGMENTS

We are thankful to J. Benskin and H. Dircksen (Stockholm University) for fruitful discussions.

This research was supported by The Swedish Research Council (VR) for BioDeg project (grant 2018-05213) and The Swedish Research Council for Sustainable Development (FORMAS) (grant 2018-01010) to E.G. and R.E.

R.E. developed the hypothesis with contribution from E.G., participated in NGS bioinformatics analysis, and wrote the first draft. E.G. contributed to writing and conducted the data analysis, including bioinformatic and statistical analyses with some contributions from R.E. R.E., S.L.-J., and A.R. conducted the laboratory experiments. A.R. conducted LC-MS/MS data analysis.

We declare that the research was conducted in the absence of any commercial or financial relationships that could be construed as a potential conflict of interest.

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
