## [Reviewer comments · mSystems]

Microbiota-dependent and independent production of L-Dopa in the gut of *Daphnia magna*

Rehab El-Shehawy, Sandra Luecke-Johansson, Anton Ribbenstedt, and Elena Gorokhova

Corresponding Author(s): Elena Gorokhova, Stockholm University

Review Timeline:

Submission Date:	August 2, 2021
Editorial Decision:	September 10, 2021
Revision Received:	October 18, 2021
Accepted:	October 19, 2021

Editor: David Cleary

Reviewer(s): Disclosure of reviewer identity is with reference to reviewer comments included in decision letter(s). The following individuals involved in review of your submission have agreed to reveal their identity: Reilly O Cooper (Reviewer #1); Ellen Decaestecker (Reviewer #2)

Transaction Report:

DOI: <https://doi.org/10.1128/mSystems.00892-21>

September 10, 2021

Prof. Elena Gorokhova
Stockholm University
Department of Environmental Science
Stockholm
Sweden

Re: mSystems00892-21 (Microbiota-dependent and independent production of L-Dopa in the gut of *Daphnia magna*)

Dear Prof. Elena Gorokhova:

Thank you for submitting your manuscript to mSystems. First, let me apologise for the length of time this initial review process has taken. We have completed our review and I am pleased to inform you that, in principle, we expect to accept it for publication in mSystems. However, acceptance will not be final until you have adequately addressed the reviewer comments.

From my perspective, I am in agreement with Reviewer 1 that additional detail should be provided on the methods used for 16S analysis. Noting the primers, variable regions used, the overlap used for merging reads, how taxonomy was assigned and whether or not you made use of the SILVA species add-on. I also wonder if you could indicate whether any controls were used in your sequencing? The issue of contamination in 16S studies as you will know are very well highlighted. May I also suggest that you deposit all R code in an appropriate repository - this is essential for reproducibility and would go some way to answering most of my queries regarding transformation of data (I note you did not rarefy, which is great, but did you examine log transformation for example?). Finally, please also add an importance section to your manuscript.

Preparing Revision Guidelines

Sincerely,

David Cleary

Editor, mSystems

Journals Department
American Society for Microbiology
1752 N St., NW

Reviewer comments:

Reviewer #1 (Comments for the Author):

In this manuscript, the authors use a well-designed set of experiments to understand the potential for cross-kingdom signaling in the model zooplankton species *Daphnia magna* and its microbiome. Generally, the manuscript is very well-written and the hypotheses, experiments, and results are clearly laid out. This is a compelling early step in understanding the underlying mechanisms of host-microbe interactions in this model system, where microbes clearly have an impact on host fitness yet very few studies have begun to explore this mechanistically. I commend the authors on a paper that reveals fascinating insights into this inter-kingdom relationship.

Comments:

There appears to be a gap in the introduction between discussing the ability of vertebrate-associated bacteria to interact with neurotransmitters and the potential for this in invertebrates, particularly in zooplankton. In particular, the authors state in the first paragraph (Lines 43-46) that bacteria recognize and produce those neurotransmitters, but then in paragraph 3 (Lines 54-56) only say that CA-mediated communication is likely to occur. I believe it would benefit the manuscript substantially to include more relevant information on invertebrate neurotransmitters, or at least how this is biologically relevant for invertebrates. For example, some work indicates that neurotransmitters induce predator responses in *Daphnia* species (Weiss et al., 2012 <https://doi.org/10.1371/journal.pone.0036879>), and Jia et al. 2021 (<https://doi.org/10.1038/s41467-021-23041-y>) indicates that *Drosophila* behavior is mediated through gut microbiome-produced neurotransmitters.

Please specify the hypervariable regions used for 16S rRNA gene sequencing (probably Line 212 would be the most relevant point to mention this). In the BioProject I was able to find it is V3-V4.

Was taxonomy classification with the SILVA database done with dada2's assignTaxonomy or with DECIPHER's IdTaxa function? Knowing this would be helpful for understanding the included phylogeny and differential abundance analysis results, as IdTaxa is generally less relaxed about identifying sequences (i.e., classifying more sequences as unknown), which would affect differential abundance analysis. Related to this, it would be greatly appreciated if a link to the code used for these analyses (GitHub, Bitbucket, Zenodo even) were available to peruse.

The statement on Lines 254-255 that "the bacterial communities obtained in the L-Tyrosine incubations were representative of the *D. magna* gut microbiome for this clone" seems to be more of a result than a method. Additionally, how was this calculated? I am extraordinarily surprised that Limnohabitans species are not indicated as present in Figure S3, as they are generally very abundant across *Daphnia magna* studies, and Comamonadaceae appear to be consistently present across cladocerans (Eckert et al., 2021, <https://doi.org/10.1111/mec.15815>). There are two technical reasons this may have occurred, though it may be entirely possible your *Daphnia* did not have Comamonadaceae: (1) the trim lengths used in your dada2 parameters are leading to inconsistent merging - truncLen = c(270, 200) is likely better than the c(240, 200) it appears you are using, and (2) the SILVA database may underrepresent Comamonadaceae sequences, though that is less likely. I would suggest re-running the dada2 analysis with an increased overlap between forward and reverse reads to see if merging indicates ASVs in line with other *Daphnia* studies.

Related to the above comment and more generally, several of the "standard" 16S rRNA analyses have been omitted. While the main focus of this paper is justifiably on the fascinating interaction between hosts and microbes with neurotransmitters, I do think a more traditional view of microbiome composition than the phylogenetic tree of Figure S3 would be useful for other zooplankton microbiome researchers to understand. This would be especially useful to understand the underlying context of the differential abundance analysis - are the highly abundant taxa the ones fluctuating, or is it rare taxa?

Several studies have indicated that DESeq2 is better adapted for microbiome studies in determining differential relative abundances (Calgaro et al. 2020 demonstrates this best in my opinion <https://doi.org/10.1186/s13059-020-02104-1>). If not inconvenient for the authors, I would strongly suggest running the differential analysis using DESeq2 to reduce some of the false positive rate.

In the discussion, the authors state that the free, unconjugated L-Dopa supports the suggested pathway. This is a wonderful result! I would appreciate if the authors could reiterate the suggested pathway in that sentence (Lines 332-334) instead of only pointing to Figure 1, as briefly reiterating the pathway could help less knowledgeable readers remember the overarching result.

In Lines 336-339, the authors discuss L-Dopa transport via a neutral amino acid transporter. Several metagenome-assembled genomes had genes for amino acid exporters of this general type in *Daphnia magna* (Cooper & Cressler 2020,

<https://doi.org/10.1038/s41598-019-57367-x>), which supports this result and could be mentioned here.

Could Ddc expression peaking after molt simply be due to size increase in Daphnia during that time? Daphnia grow substantially during these instars, so I am wondering if any size correction was utilized to account for this. Alternatively, is it also possible that this may coincide with increases in bacterial abundance? There is more surface area available both as Daphnia grow and as molting occurs, which allows more bacteria to be present. This is not necessarily something you were able to answer with these carefully crafted experiments, but I am wondering if it could be proposed as alternative hypotheses for unregulated Ddc.

The "The" as the first word of the abstract could be removed.

It appears that "dopa" is capitalized inconsistently throughout the manuscript (dopa decarboxylase, Line 21; DOPA decarboxylase, Line 21). Is that to denote different uses? If not, unifying the capitalization would be helpful.

Reviewer #2 (Comments for the Author):

Dear authors, I have read with great interest your study, which I think is highly innovative. So far, no one has addressed the role of metabolites in Daphnia-microbiome research, and this is a very good attempt with multiple and sound techniques, confirming the findings. The approach and results are sound. I don't have major objections. My minor comments are: i) if you would use multiple Daphnia genotypes would your results be consistent? please address this in the discussion; ii) only one reference gene is used in the RT-qPCR. Please address this as well in the discussion; multiple reference genes are recommended in such gene expression studies.

Microbiota-dependent and independent production of L-dopa in the gut of *Daphnia magna*

#paperreview

Rehab El-Shehawy et al.

mSystems

In this manuscript, the authors use a well-designed set of experiments to understand the potential for cross-kingdom signaling in the model zooplankton species *Daphnia magna* and its microbiome. Generally, the manuscript is very well-written and the hypotheses, experiments, and results are clearly laid out. This is a compelling early step in understanding the underlying mechanisms of host-microbe interactions in this model system, where microbes clearly have an impact on host fitness yet very few studies have begun to explore this mechanistically. I commend the authors on a paper that reveals fascinating insights into this inter-kingdom relationship.

Comments:

There appears to be a gap in the introduction between discussing the ability of vertebrate-associated bacteria to interact with neurotransmitters and the potential for this in invertebrates, particularly in zooplankton. In particular, the authors state in the first paragraph (Lines 43-46) that bacteria recognize and produce those neurotransmitters, but then in paragraph 3 (Lines 54-56) only say that CA-mediated communication is likely to occur. I believe it would benefit the manuscript substantially to include more relevant information on invertebrate neurotransmitters, or at least how this is biologically relevant for invertebrates. For example, some work indicates that neurotransmitters induce predator responses in *Daphnia* species (Weiss et al., 2012 <https://doi.org/10.1371/journal.pone.0036879>), and Jia et al. 2021 (<https://doi.org/10.1038/s41467-021-23041-y>) indicates that *Drosophila* behavior is mediated through gut microbiome-produced neurotransmitters.

Please specify the hypervariable regions used for 16S rRNA gene sequencing (probably Line 212 would be the most relevant point to mention this). In the BioProject I was able to find it is V3-V4.

Was taxonomy classification with the SILVA database done with dada2's assignTaxonomy

or with DECIPHER's IdTaxa function? Knowing this would be helpful for understanding the included phylogeny and differential abundance analysis results, as IdTaxa is generally less relaxed about identifying sequences (i.e., classifying more sequences as unknown), which would affect differential abundance analysis. Related to this, it would be greatly appreciated if a link to the code used for these analyses (GitHub, Bitbucket, Zenodo even) were available to peruse.

The statement on Lines 254-255 that "the bacterial communities obtained in the L-Tyrosine incubations were representative of the *D. magna* gut microbiome for this clone" seems to be more of a result than a method. Additionally, how was this calculated? I am extraordinarily surprised that Limnohabitans species are not indicated as present in Figure S3, as they are generally very abundant across *Daphnia magna* studies, and Comamonadaceae appear to be consistently present across cladocerans (Eckert et al., 2021, <https://doi.org/10.1111/mec.15815>). There are two technical reasons this may have occurred, though it may be entirely possible your *Daphnia* did not have Comamonadaceae: (1) the trim lengths used in your dada2 parameters are leading to inconsistent merging - `truncLen = c(270, 200)` is likely better than the `c(240, 200)` it appears you are using, and (2) the SILVA database may underrepresent Comamonadaceae sequences, though that is less likely. I would suggest re-rerunning the dada2 analysis with an increased overlap between forward and reverse reads to see if merging indicates more general ASVs in line with other *Daphnia* studies.

Related to the above comment and more generally, several of the "standard" 16S rRNA analyses have been omitted. While the main focus of this paper is justifiably on the fascinating interaction between hosts and microbes with neurotransmitters, I do think a more traditional view of microbiome composition than the phylogenetic tree of Figure S3 would be useful for other zooplankton microbiome researchers to understand. This would be especially useful to understand the underlying context of the differential abundance analysis - are the highly abundant taxa the ones fluctuating, or is it rare taxa?

Several studies have indicated that DESeq2 is better adapted for microbiome studies in determining differential relative abundances (Calgaro et al. 2020 demonstrates this best in my opinion <https://doi.org/10.1186/s13059-020-02104-1>). If not inconvenient for the authors, I would strongly suggest running the differential analysis using DESeq2 to reduce some of the false positive rate.

In the discussion, the authors state that the free, unconjugated L-Dopa supports the suggested pathway. This is a wonderful result! I would appreciate if the authors could reiterate the suggested pathway in that sentence (Lines 332-334) instead of only pointing

to Figure 1, as briefly reiterating the pathway could help less knowledgeable readers remember the overarching result.

In Lines 336-339, the authors discuss L-Dopa transport via a neutral amino acid transporter. Several metagenome-assembled genomes had genes for amino acid exporters of this general type in *Daphnia magna* (Cooper & Cressler 2020, <https://doi.org/10.1038/s41598-019-57367-x>), which supports this result and could be mentioned here.

Could *Ddc* expression peaking after molt simply be due to size increase in *Daphnia* during that time? *Daphnia* grow substantially during these instars, so I am wondering if any size correction was utilized to account for this. Alternatively, is it also possible that this may coincide with increases in bacterial abundance? There is more surface area available both as *Daphnia* grow and as molting occurs, which allows more bacteria to be present. This is not necessarily something you were able to answer with these carefully crafted experiments, but I am wondering if it could be proposed as alternative hypotheses for unregulated *Ddc*.

The "The" as the first word of the abstract could be removed.

It appears that "dopa" is capitalized inconsistently throughout the manuscript (dopa decarboxylase, Line 21; DOPA decarboxylase, Line 21). Is that to denote different uses? If not, unifying the capitalization would be helpful.

Dear Editor and Reviewers,

Thank you very much for taking the time to read our manuscript, asking relevant questions, and suggesting additional references. We have considered all your comments, which helped improve the manuscript, especially the NGS data presentation. Below you will find our replies numbered consequently to facilitate cross-referencing; the line numbers in the replies refer to the revised manuscript.

Sincerely,

Elena Gorokhova

Editor:

From my perspective, I am in agreement with Reviewer 1 that additional detail should be provided on the methods used for 16S analysis. Noting the primers, variable regions used, the overlap used for merging reads, how taxonomy was assigned and whether or not you made use of the SILVA species add-on.

Reply 1: The parts related to the 16S rRNA gene sequence analysis and data treatment methods were revised to clarify these points and provide the requested information. (Lines 224-235).

I also wonder if you could indicate whether any controls were used in your sequencing? The issue of contamination in 16S studies as you will know are very well highlighted.

Reply 2: Yes, we are aware of this issue, and *Pseudomonas* is indeed one of the species reported to occur due to contamination (Salter et al., 2014). We use no-template DNA extractions for all kits that are used for the first time in our lab. Moreover, LC Sciences / LC Bio, the company performing NGS, conducted PCR with negative controls (ultrapure water instead of DNA template) as a part of their routine protocol. None of these regular blanks (=negative controls) yielded any measurable DNA quantities; this information is now added to M&M (Lines 226-227). According to LC Sciences personnel, running no-DNA samples or non-detects after the PCR step that would contain only primers and tags can adversely affect the overall outcome of the sequencing and complicate the downstream analyses. Therefore, no sequencing for these samples was conducted, which means that the comment raises a valid issue because no blanks were sequenced, and no mock communities as positive controls were used.

However, we believe that measurable contamination is unlikely to occur in these samples, as also suggested by the negative PCR blanks. First, the sequence contamination is usually detectable only in low-template samples (Salter et al., 2014; Karstens et al., 2019), which was not the case here. In our case, all samples were bacterial cultures with high density (Fig. S1; Supplementary Materials) and high-yield DNA. Second, we also ran a pilot experiment a few months before the main experiment, where only one L-Tyr concentration (2 mM) vs. control (0 mM) was tested, and the dominance of *Pseudomonas* spp. under the L-Tyrosine enrichment was found. In this pilot run, the test samples were unreplicated, and the entire concentration range for L-Tyr was not included, which is why the result was not reported in the submitted manuscript. Nevertheless, this independent experiment confirmed that significant upregulation of *Pseudomonas* presented in the manuscript is not an artifact or the

result of contamination. Finally, the negative (ultrapure water) controls had no growth on either LB or Tyrosine, which implies that contamination during the experiment was non-detectable.

May I also suggest that you deposit all R code in an appropriate repository - this is essential for reproducibility and would go some way to answering most of my queries regarding transformation of data (I note you did not rarefy, which is great, but did you examine log transformation for example?).

Reply 3: The code for the workflow generating amplicon sequence variant (ASV) table and assigning taxonomy to the output sequences is a proprietary product of the company providing the NGS services (LC Sciences; <https://lcsciences.com/>); however, the settings applied for trimming and filtering were discussed with the bioinformatics experts of the company and are provided in the revised manuscript (Lines 226-236).

Indeed, the data were not rarefied but for the differential abundance analysis with edgeR they were scaled to TMM as recommended (Robinson and Oshlack, 2010). This information is now added to the manuscript (Lines 258-259).

Reviewer #1

There appears to be a gap in the introduction between discussing the ability of vertebrate-associated bacteria to interact with neurotransmitters and the potential for this in invertebrates, particularly in zooplankton. In particular, the authors state in the first paragraph (Lines 43-46) that bacteria recognize and produce those neurotransmitters, but then in paragraph 3 (Lines 54-56) only say that CA-mediated communication is likely to occur. I believe it would benefit the manuscript substantially to include more relevant information on invertebrate neurotransmitters, or at least how this is biologically relevant for invertebrates. For example, some work indicates that neurotransmitters induce predator responses in *Daphnia* species (Weiss et al., 2012 <https://doi.org/10.1371/journal.pone.0036879>), and Jia et al. 2021 (<https://doi.org/10.1038/s41467-021-23041-y>) indicate that *Drosophila* behavior is mediated through gut microbiome-produced neurotransmitters.

Reply 4: Good point, the gap is now filled providing information and relevant references regarding invertebrate-associated bacterial microbiome and if/how gut microbiome may modulate invertebrate neurotransmitters. (Lines 49-61).

Thank you for pointing out the work of Jia et al. 2021, which is now cited as an example of the microbiome involvement to neurotransmitter production and functioning in invertebrates. The work of Weiss et al. from 2012 (Weiss et al., 2012) as well as their following work on Dopamine-induced morphological defenses in *Daphnia* (Weiss et al., 2015) is indeed exciting and the latter was cited in the Discussion with regard to Dopamine roles in *Daphnia*. However, it does not consider any microbiome involvement; therefore, it was not exactly the match for the Introduction part.

Please specify the hypervariable regions used for 16S rRNA gene sequencing (probably Line 212 would be the most relevant point to mention this). In the BioProject I was able to find it is V3-V4. Was taxonomy classification with the SILVA database done with dada2's assign Taxonomy or with DECIPHER's IdTaxa function? Knowing this would be helpful for understanding the included phylogeny and differential abundance analysis results, as IdTaxa is generally less relaxed about

identifying sequences (i.e., classifying more sequences as unknown), which would affect differential abundance analysis.

Reply 5: The hypervariable region was V3-V4, and taxonomy was assigned using QIIME 2's q2-feature-classifier plugin (Bokulich et al., 2018) allowing for 3 mismatches between the forward and reverse sequences. We have extensively revised the description of the workflow; see **Reply 3**.

Related to this, it would be greatly appreciated if a link to the code used for these analyses (GitHub, Bitbucket, Zenodo even) were available to peruse.

Reply 6: Unfortunately, this is not possible due to the service company regulations; see **Reply 3**.

The statement on Lines 254-255 that "the bacterial communities obtained in the L-Tyrosine incubations were representative of the *D. magna* gut microbiome for this clone" seems to be more of a result than a method. Additionally, how was this calculated?

Reply 7: We agree that this might be confusing, this information is now moved to Results (Lines 304-305). By *representative*, we meant that the taxa lists for the cultivated bacteria inoculated from the daphnia gut were partially overlapping with the *Daphnia* microbiota reported for this specific clone in our recent studies (Motiei et al., 2020; Gorokhova et al., 2021). No specific calculations were attempted to demonstrate this because the exact scale of the overlap is not relevant considering that we are dealing with the culturable part of the microbiome here; see **Reply 8**.

I am extraordinarily surprised that Limnohabitans species are not indicated as present in Figure S3, as they are generally very abundant across *Daphnia magna* studies, and Comamonadaceae appear to be consistently present across cladocerans (Eckert et al., 2021, <https://doi.org/10.1111/mec.15815>). There are two technical reasons this may have occurred, though it may be entirely possible your *Daphnia* did not have Comamonadaceae: (1) the trim lengths used in your dada2 parameters are leading to inconsistent merging - truncLen = c(270, 200) is likely better than the c(240, 200) it appears you are using, and (2) the SILVA database may underrepresent Comamonadaceae sequences, though that is less likely. I would suggest re-rerunning the dada2 analysis with an increased overlap between forward and reverse reads to see if merging indicates ASVs in line with other *Daphnia* studies.

Reply 8: We agree that both Limnohabitans and Comamonadaceae are regular members of *Daphnia* microbiome and this is also true for the gut microbiome of our clone identified using culture-free NGS (Motiei et al., 2020; Gorokhova et al., 2021). However, the data reported here are for the gut community that was first used as inoculum for a culture grown in LB medium (to enrich the community in culturable taxa), which in turn was used as inoculum in treatments with L-Tyrosine (to enrich the community in taxa that can use Tyrosine as a substrate). These manipulations resulted in the selective growth of some community members and the disappearance of others, and it would be unreasonable to expect the dominance structure to be preserved in the samples generated by the *in vitro* exposure. This is in line with our recent study (Gorokhova et al., 2021), where the daphnid microbiome was used to inoculate the experimental system consisting of water and suspended solids. After 4 days of exposure (i.e., approximately the same time as in the current study), the bacterial community originated from the microbiome and comprising the biofilm associated with the solids was drastically different from the original microbiome, albeit the taxa lists were largely

overlapping (see Fig. 4 in Gorokhova et al., 2021). Thus, this was a similar and expected outcome resulting from the differential selection of microorganisms in vitro conditions.

Related to the above comment and more generally, several of the "standard" 16S rRNA analyses have been omitted. While the main focus of this paper is justifiably on the fascinating interaction between hosts and microbes with neurotransmitters, I do think a more traditional view of microbiome composition than the phylogenetic tree of Figure S3 would be useful for other zooplankton microbiome researchers to understand. This would be especially useful to understand the underlying context of the differential abundance analysis - are the highly abundant taxa the ones fluctuating, or is it rare taxa?

Reply 9: Of course, we can add more of the "standard" plots and tests, but we have doubts that they would be valuable for comparisons with *Daphnia* microbiome *per se* because these communities were the result of LB and L-Tyrosine exposure and, therefore, they were much different (in terms of the dominance structure) from the original daphnid microbiome; see **Reply 8**. Therefore, none of these communities would be comparable in terms of relative abundance with the original gut microbiome that was used for inoculation. We would like to stress that the goal of this experiment was to provide evidence that the gut microbiota contains taxa capable of L-Tyrosine utilization and L-Dopa production, and to find which taxa were associated with this production. The taxa being abundant are those responding best to the enrichment with LB and when provided a surplus of L-Tyrosine; see **Replies 7 and 8**.

Several studies have indicated that DESeq2 is better adapted for microbiome studies in determining differential relative abundances (Calgaro et al. 2020 demonstrates this best in my opinion <https://doi.org/10.1186/s13059-020-02104-1>). If not inconvenient for the authors, I would strongly suggest running the differential analysis using DESeq2 to reduce some of the false positive rate.

Reply 10: When preparing the manuscript, we did use both DESeq2 and edgeR for the differential abundance analysis and the overall conclusion was largely the same, although DESeq2 had indeed provided lower log2FC value due to the fact that it uses Foldchange-shrinkage. Here is the comparison for the output between the two methods:

Method	log2FC	logCPM	Pvalues	FDR
edgeR	1.3809	20.174	0.001652	0.009165
DEseq2	1.2811	7.8454	0.010788	0.035211

The important thing here is that both models capture the reality of the biological data shown in Fig. 4 and not perhaps how close to each other log2FC and, especially, the p values are. In this case, the data scatter visualized in the figure is in line with the test output. Please, observe that at this stage *Pseudomonas* upregulation is only a correlative observation, not the causal evidence for the involvement of these bacteria in L-Dopa production.

In the discussion, the authors state that the free, unconjugated L-Dopa supports the suggested pathway. This is a wonderful result! I would appreciate if the authors could reiterate the suggested pathway in that sentence (Lines 332-334) instead of only pointing to Figure 1, as briefly reiterating the pathway could help less knowledgeable readers remember the overarching result.

Reply 11: Thank you, done (Lines 338-341).

In Lines 336-339, the authors discuss L-Dopa transport via a neutral amino acid transporter. Several

metagenome-assembled genomes had genes for amino acid exporters of this general type in *Daphnia magna* (Cooper & Cressler 2020, <https://doi.org/10.1038/s41598-019-57367-x>), which supports this result and could be mentioned here.

Reply 12: Thank you, very good point. We have now included this information and the reference in the Discussion (Lines 342-349).

Could *Ddc* expression peaking after molt simply be due to size increase in *Daphnia* during that time? *Daphnia* grow substantially during these instars, so I am wondering if any size correction was utilized to account for this.

Reply 13: We always load an equal amount of DNA in qPCR reactions; therefore, the absolute expression would to some extent represent mass-specific values of the *Ddc* expression in the gut tissues. Moreover, the housekeeping gene would have compensated for any size increase in *Daphnia* gut after molt. Therefore, we would consider the size (i.e., individual mass) effect unlikely.

Alternatively, is it also possible that this may coincide with increases in bacterial abundance? There is more surface area available both as *Daphnia* grow and as molting occurs, which allows more bacteria to be present. This is not necessarily something you were able to answer with these carefully crafted experiments, but I am wondering if it could be proposed as alternative hypotheses for unregulated *Ddc*.

Reply 14: Please, observe that the *Ddc* primers are *Daphnia*-specific, so they should not amplify *Ddc* gene in prokaryotes. To confirm that no non-specific amplification were produced, we have tested the primers against bacterial DNA. Moreover, the melt curves were also examined in the qPCR runs and no non-specific peaks were observed. So, we are very confident that the gene expression data were host-specific.

The "The" as the first word of the abstract could be removed.

Reply 15: Done.

It appears that "dopa" is capitalized inconsistently throughout the manuscript (dopa decarboxylase, Line 21; DOPA decarboxylase, Line 21). Is that to denote different uses? If not, unifying the capitalization would be helpful.

Reply 16: Unified to dopa decarboxylase (the enzyme) and L-Dopa (the molecule). Thank you.

Reviewer #2:

i) if you would use multiple *Daphnia* genotypes would your results be consistent? please address this in the discussion;

Reply 17: This is a valid question for future work given the observed variability in microbiome between clones and species; pointed out in the Discussion (Lines 396-398).

ii) only one reference gene is used in the RT-qPCR. Please address this as well in the discussion; multiple reference genes are recommended in such gene expression studies.

Reply 18: We have tested G3PDH and RNAPII as the housekeeping genes, both were demonstrated to have stable expression in *Daphnia* (Scoville and Pfrender, 2010, Heckman et al., 2006). Whereas RNAPII had some variability, G3PDH was more stable (also noted by Heckman and co-workers) and not affected over the molting cycle, which was the critical point in our experiment. Therefore, we decided to use only the latter for the ΔC_t calculations.

References

- Bokulich, N. A., Kaehler, B. D., Rideout, J. R., Dillon, M., Bolyen, E., Knight, R., et al. (2018). Optimizing taxonomic classification of marker-gene amplicon sequences with QIIME 2's q2-feature-classifier plugin. *Microbiome* 6, 90. doi:10.1186/s40168-018-0470-z.
- Gorokhova, E., Motiei, A., and El-Shehawy, R. (2021). Understanding Biofilm Formation in Ecotoxicological Assays With Natural and Anthropogenic Particulates. *Front. Microbiol.* 12. doi:10.3389/fmicb.2021.632947.
- Karstens, L., Asquith, M., Davin, S., Fair, D., Gregory, W. T., Wolfe, A. J., et al. (2019). Controlling for Contaminants in Low-Biomass 16S rRNA Gene Sequencing Experiments. *mSystems* 4, e00290-19. doi:10.1128/mSystems.00290-19.
- Motiei, A., Brindefalk, B., Ogonowski, M., El-Shehawy, R., Pastuszek, P., Ek, K., et al. (2020). Disparate effects of antibiotic-induced microbiome change and enhanced fitness in *Daphnia magna*. *PLOS ONE* 15, e0214833. doi:10.1371/journal.pone.0214833.
- Robinson, M. D., and Oshlack, A. (2010). A scaling normalization method for differential expression analysis of RNA-seq data. *Genome Biology* 11, R25. doi:10.1186/gb-2010-11-3-r25.
- Salter, S. J., Cox, M. J., Turek, E. M., Calus, S. T., Cookson, W. O., Moffatt, M. F., et al. (2014). Reagent and laboratory contamination can critically impact sequence-based microbiome analyses. *BMC Biology* 12, 87. doi:10.1186/s12915-014-0087-z.
- Scoville, A. G., and Pfrender, M. E. (2010). Phenotypic plasticity facilitates recurrent rapid adaptation to introduced predators. *PNAS* 107, 4260–4263. doi:10.1073/pnas.0912748107.
- Weiss, L. C., Kruppert, S., Laforsch, C., and Tollrian, R. (2012). Chaoborus and Gasterosteus Anti-Predator Responses in *Daphnia pulex* Are Mediated by Independent Cholinergic and Gabaergic Neuronal Signals. *PLOS ONE* 7, e36879. doi:10.1371/journal.pone.0036879.
- Weiss, L. C., Leese, F., Laforsch, C., and Tollrian, R. (2015). Dopamine is a key regulator in the signalling pathway underlying predator-induced defences in *Daphnia*. *Proceedings of the Royal Society B: Biological Sciences* 282, 20151440. doi:10.1098/rspb.2015.1440.

October 19, 2021

Prof. Elena Gorokhova
Stockholm University
Department of Environmental Science
Stockholm
Sweden

Re: mSystems00892-21R1 (Microbiota-dependent and independent production of L-Dopa in the gut of *Daphnia magna*)

Dear Prof. Elena Gorokhova:

Thank you for submitting the revisions. I am pleased to tell you that your manuscript has been accepted, and I am forwarding it to the ASM Journals Department for publication. For your reference, ASM Journals' address is given below. Before it can be scheduled for publication, your manuscript will be checked by the mSystems senior production editor, Ellie Ghatineh, to make sure that all elements meet the technical requirements for publication. She will contact you if anything needs to be revised before copyediting and production can begin. Otherwise, you will be notified when your proofs are ready to be viewed.

As an open-access publication, mSystems receives no financial support from paid subscriptions and depends on authors' prompt payment of publication fees as soon as their articles are accepted. =

Publication Fees:

We recognize that the video files can become quite large, and so to avoid quality loss ASM suggests sending the video file via <https://www.wetransfer.com/>. When you have a final version of the video and the still ready to share, please send it to Ellie Ghatineh at eghatineh@asmusa.org.

Sincerely,

David Cleary
Editor, mSystems

Journals Department
Fig. S4: Accept
Fig. S3: Accept
Table S1: Accept
Fig. S2: Accept
Fig. S1: Accept